# The Missing Measure of Loneliness: A Case for Including Neededness in Loneliness Scales

**DOI:** 10.3390/ijerph19010429

**Published:** 2021-12-31

**Authors:** Ariel Gordy, Helen Han Wei Luo, Margo Sidline, Kimberley Brownlee

**Affiliations:** Philosophy Department, University of British Columbia, Vancouver, BC V6T 1Z1, Canada; asjgubc@student.ubc.ca (A.G.); hwluo@student.ubc.ca (H.H.W.L.); msidline@student.ubc.ca (M.S.)

**Keywords:** belonging, De Jong-Gierveld Scale, loneliness, loneliness scales, neededness, social connection, social needs, UCLA Loneliness Scale

## Abstract

Prominent tools used to measure loneliness such as the UCLA Scale and DJGS include no items related to being needed, i.e., *neededness*. More recent scales such as the DLS and SELSA do include items on neededness, but only within their romantic loneliness subscales. This paper proposes that new iterations of loneliness scales should include in all subscales two items on neededness: (a) whether a person feels important to someone else and (b) whether that person has good ways to serve others’ well-being. The paper surveys cognate studies that do not rely on loneliness scales but establish a link between neededness and feelings of social connection. It then highlights ways in which neededness items would improve the ability of loneliness scales to specify the risk profile, to delineate variations in the emotional tone and quality of loneliness, and to propose suitable interventions. The paper outlines a theoretical argument—drawing on moral philosophy—that prosociality and being needed are non-contingent, morally urgent human needs, postulating that the protective benefits of neededness vary according to at least four factors: the significance, persistence, non-instrumentality, and non-fungibility of the ways in which a person is needed. Finally, the paper considers implications for the design of appropriate remedies for loneliness.

## 1. Introduction

A growing body of social psychology evidence identifies prosocial behaviour and feelings of ‘being needed’ as protective factors for well-being [1,2]. A subset of this evidence suggests that specific factors—referred to here as *neededness* factors—including (a) feeling important to someone else and (b) being useful to others, are key to mitigating loneliness [3,4]. When it comes to measuring loneliness, however, prominent tools such as the UCLA Loneliness Scale leave out such neededness factors. This raises difficulties because loneliness scales are used both to isolate the precursors that tend to lead to loneliness and to specify the nature, tone, and severity of loneliness. If a person’s loneliness is due to feeling unneeded or unable to further others’ well-being, this will go largely undetected in existing loneliness scales. Not only does this produce an incomplete, if not inaccurate, picture of the experience of loneliness, but it also has deleterious implications for a person’s prospects for treatment and remedy.

This paper proposes that new iterations of loneliness scales should incorporate two kinds of neededness items in order to study accurately the degree of association between impoverished social contribution and loneliness, with one item focusing on whether a person perceives themself as important to, or needed by, one or more others, and one item focusing on whether they perceive that they have good ways to contribute to others’ well-being. 

This paper provides an overview of the development and content of existing loneliness scales (Section 2); reviews a selection of cognate studies that identify a link between prosociality, utility, and well-being (Section 3); summarises the findings in some qualitative studies in which lonely people report feeling unneeded (Section 4); notes ways that loneliness scales would be improved through the inclusion of neededness items (Section 5); outlines the theoretical case for the view that prosociality and neededness (of a specific standard) are non-contingent, fundamental human needs, a view that has implications for public policy (Section 6); and, finally, considers some implications of taking neededness as a key reference point in the design of treatments and remedies for loneliness, assessing the adequacy of different social contexts and association types as better or worse sites in which to achieve adequate levels of neededness (Section 7). 

Before proceeding, it is necessary to be clear on what is meant by the concept of *need* in the context of need fulfilment and neededness. In its most minimal form, a *need* is anything that is necessary to achieve an end, which means that not all needs are weighty or even morally acceptable. To understand the nature and force of a given need, one must ask the question: ‘What for’? [5]. If a person says they need air, the reason might be that they intend to blow up some balloons with it or that they need air to breathe. If they say they need water, the reason might be that they wish to wash dishes or they need it to drink or, indeed, they intend to drown an insect. These examples point to an important distinction between *contingent needs* and *non-contingent needs*, which will be discussed in Section 6. Whereas contingent needs are dependent on changeable conditions such as a person’s desires, non-contingent needs are independent of such conditions: they are the needs inherent to a being by virtue of its existence as the kind of being it is [5,6]. For instance, a person not only needs air to breathe and water to drink, but also needs sufficient, nourishing food to eat irrespective of their desires to live, drink, or eat. As Section 6 will indicate, neededness is arguably another non-contingent need that human beings cannot help but have.

A potential ambiguity concerning the concept of *neededness* is whether it is an objective measure (of how much a person is actually valued by others and useful to others’ well-being) or a subjective measure (of a person’s perception of how much they are valued by and useful to others). The theoretical account outlined in Section 6 argues that neededness in the objective sense is a non-contingent human need. But of course, the measure that this paper proposes for inclusion in loneliness scales is a subjective measure of a person’s perceptions of their value to others. If a person’s subjective perceptions that they are needed are indeed indispensable to their well-being, then those perceptions too might constitute a non-contingent need. However, even if we bracket the fact that it would be remarkably challenging to meet such perceptual needs, a difficulty remains. When a person’s perceptions that they are needed are ill-founded, those perceptions can give that person a false sense of social security that could ultimately threaten their well-being. One task for future psychological work is to determine to what extent the objective and subjective measures of neededness are independent of each other or interlinked and mutually reinforcing. It is certainly possible that a person could be greatly needed by others but perceive themself as not needed, such as a person who is experiencing clinical depression or suicidal ideation. It is also possible that a person might not be needed much at all by others, but perceive themself as needed, such as a busybody or someone exhibiting features of a ‘white savior complex’.

## 2. Overview of Existing Loneliness Scales

Transient loneliness is a common human experience that is markedly distinct from chronic loneliness, which occurs when a person experiences feelings of loneliness long enough to develop a ‘persistent, self-reinforcing loop of negative thoughts, sensations, and behaviours’ [7] (p. 7). Transient loneliness is a protective evolutionary mechanism that alerts a person to potential gaps or inadequacies in their social networks in the same way that physical pain, hunger, and thirst alert a person to bodily threats. Chronic loneliness, by contrast, can pose a serious threat to a person’s health and well-being; it is correlated with a host of health risks, including reduced immunity, progression of Alzheimer’s disease, alcoholism, stroke, depression, suicidal ideation, and suicidal behaviour [8,9]. It is described in some studies as worse for health than smoking 15 cigarettes a day [10].

In academic research, an array of scales now exists that aim to measure the experience of loneliness. One of the first scales to be widely accepted is the UCLA Loneliness Scale. This 20-item scale was developed in 1978 out of a need for a reliable loneliness measure and as a tool to analyse the impact that loneliness can have on health [9]. This scale assesses the frequency of feelings of loneliness by grading items such as ‘How often do you feel alone?’ and ‘How often do you feel left out?’ on a four-point scale ranging from ‘never’ to ‘always’ (later revised to ‘often’) [11]. Shortly after the UCLA Scale was developed, Jenny De Jong-Gierveld et al. formulated a new loneliness measure in 1985, the De Jong-Gierveld Scale (DJGS), which also aims to assess the frequency of feelings of deprivation that accompany loneliness. The DJGS consists of 11 items, and it too has been widely accepted in loneliness research as a reliable measure [12]. However, as more researchers have turned their attention to loneliness, a consensus has arisen that something is missing in these scales, following Robert Weiss’s observation that loneliness is not a unidimensional experience that varies only in intensity and frequency, but rather a multidimensional experience. Weiss pointed out that there is a substantial difference between what he called *emotional loneliness* and *social loneliness*: emotional loneliness is the result of a lack of close interpersonal relationships, while social loneliness is the result of an inadequate social network [13] (pp. 18–20). In other words, the loneliness experienced by someone who just moved to a new town is an entirely different dimension of loneliness from that experienced by someone who just lost his significant other. Since these two states of loneliness have different precursors, are associated with different feelings, and have different outcomes, most researchers have embraced the multidimensional picture of loneliness that distinguishes them [14] (p. 128). (The terminology of ‘emotional loneliness’ and ‘social loneliness’ poorly captures the fact that a person may feel lonely in one sphere of their life but not in another, because a person’s experience of loneliness in any sphere of their life will presumably have an *emotional* element to it. Moreover, all (or almost all) of the settings in which a person might experience loneliness—from their intimate family setting to their workplace or their broader community—are *social* spheres. Nonetheless, we use Weiss’s terminology here because the distinction it identifies is a well-established one in the psychology literature.)

In response to this new dichotomy between social and emotional loneliness, researchers have sought to develop new loneliness measures that assess loneliness across multiple relationships, such as the Differential Loneliness Scale (DLS) and the Social and Emotional Loneliness Scale for Adults (SELSA). Both scales incorporate the multidimensional nature of loneliness into their measurements through the use of subscales. The subscales assess loneliness across different kinds of relationships, including family, romantic, and social relationships. The DLS goes further and adds a broader community subscale to measure the loneliness that a person can experience if they feel out of place in society [15] (pp. 1042–1043). The items in these scales are modified to assess different relationships with the aim of isolating the underlying causes of loneliness and to ask questions related to feelings and relationship dynamics. Items in the SELSA consist of prompts such as: ‘I feel alone when I’m with my family’, and ‘I have a lot in common with others’ [14] (p. 130). Items in the DLS include: ‘Most of my friends understand my motives and reasoning’ and ‘I work well with others in a group’ [15] (p. 1042). In general, multidimensional measures have gained wide acceptance in loneliness research as psychologists have come to appreciate how people can experience a specific sense of loneliness through their different relationships. These scales have become pivotal in identifying not only the intensity or frequency of a person’s loneliness, but the foundations of loneliness as well.

Research validating the UCLA Scale has found that it is also able to assess loneliness across multiple dimensions. For instance, Enrico DiTommaso and Barry Spinner found that all three subscales on the SELSA are significantly correlated to the UCLA Scale, with the strongest correlation existing between the social subscale of the SELSA and the UCLA Scale [14] (p. 131). Similarly, a 1996 study by Daniel Russell found that the scores on the UCLA Scale are also strongly correlated with the Differential Loneliness Scale [16] (p. 27). A more recent study by DiTomasso et al. (2004) concluded that the correlations between the scores on the UCLA Scale and the SELSA are so significant that this could indicate that the UCLA Scale displays a similar capacity to measure multidimensional loneliness experiences [17] (p. 110). This inference was more thoroughly explored in a 2005 study by Louise Hawkley et al., where they grouped UCLA Scale items into subscales to reflect a ‘three-dimensional conceptual structure’. These dimensions are classified as (1) isolation, (2) relational connectedness, and (3) collective connectedness. These subscales are designed to assess feelings of loneliness on the individual, relational, and collective levels [18] (p. 789, 799). These studies indicate that the UCLA Scale is also an effective tool in measuring experiences of loneliness across multiple relational dimensions.

As part of a growing acknowledgement that prosociality is protective of well-being and, thereby, possibly a protection against loneliness, the DLS and SELSA take one step toward incorporating into their measures both contribution-related items and items pertaining to a person’s sense of importance to others. In both scales, however, these neededness items are restricted to the romantic subscales. There are three contribution-related and importance-related items in the romantic subscale of the SELSA: ‘I am an important part of someone else’s life’, ‘There is someone else who wants to share their life with me’, and ‘I have a romantic partner to whose happiness I contribute’ [14] (p. 130). In the DLS, there are two items that take a similar form: ‘I am now involved in a romantic or marital relationship where both of us make a genuine effort at cooperation’ and ‘I am an important part of the emotional and physical well-being of my lover or spouse’ [15] (p. 1042). By incorporating these items into their multidimensional loneliness scales, the designers signal that they are aware that a person’s ability to contribute and their sense of importance to someone else can impact upon their experience of loneliness. However, the designers overlook the fact that neededness items are also highly relevant outside of the romantic context. Such items are relevant to understanding not only a person’s perceptions of their family relations including their sense of their place as a parent, grandparent, sibling, young child, adolescent, or adult child within a family, but also their perceptions of the social-contribution opportunities and connections that their friendships, workplace, or broader community can supply.

Questions pertaining to neededness are particularly important to understanding the experiences of a person who lacks a romantic partner or any close family ties. One hypothesis to test is whether such a non-attached person could achieve adequate levels of neededness within either a work environment or the broader community, and whether doing so might protect that person from loneliness.

Questions pertaining to neededness are also highly relevant to the study of specific demographics’ experience of loneliness, including notably the experiences of children and adolescents. Comparatively few studies have been conducted to date on young persons’ experiences of loneliness. Those studies that do focus on young people tend to collapse the distinction between felt loneliness and actual social rejection. A study by Pamela Qualter and Penny Munn (2002) disambiguates the two and indicates that while some children do experience both feelings of loneliness and actual social rejection, it is more common for children to experience one or the other but not both, and Qualter and Munn’s results indicate that it is loneliness and not rejection that co-occurs with emotional problems [19]. Studies such as this are noteworthy in the context of neededness because they imply that the *subjective* measure of neededness is more relevant than the objective measure is to the study of children’s social and emotional well-being.

The proposal of this paper is that loneliness scales should include neededness items in the subscales pertaining to friendship, family, and community in order to assess a person’s feelings of utility and importance across different relationships. The next section offers a first line of support for this proposal by reviewing a selection of studies that use methods other than loneliness scales to identify links between prosociality, social connectedness, and well-being.

## 3. Review of Selected Studies on Prosociality, Social Connectedness, and Well-Being

In addition to his analysis of social and emotional loneliness, Weiss postulates that deficiencies in close relationships and social networks are the root causes of loneliness. These deficiencies, according to Weiss, consist of a lack of social provisions [13] (p. 17). He identifies six social provisions that can protect a person from loneliness, namely, attachment, social integration, reliable alliance, guidance, reassurance of worth, and opportunity for nurturance [20]. Since the publication of Weiss’s work, a growing body of research has shown that contribution to others’ well-being is also a protective factor that can obviate or lessen loneliness. For example, a 2013 study by Eva Kahana et al. indicated that engaging in altruistic behaviour (i.e., selfless behaviour that aims to sustain others) is a significant predictor of positive affect in older American adults [21] (p. 159). Likewise, a 2017 study by Juan Xi et al. found that altruistic behaviours are intrinsically rewarding to the people who perform them and directly impact feelings of connectedness by increasing personal bonds [22] (pp. 71, 80). Feelings of increased connectedness contribute in turn to overall well-being in the participants performing altruistic behaviours. Relatedly, according to a 2009 study by David Mellor et al., contributing through volunteer work is positively correlated with perceived well-being and optimism, suggesting that there exists a fundamental link between prosocial behaviour or altruism and well-being in terms of psychological-need fulfilment [23]. This reward system of altruism and well-being was further analysed in a 2018 study by Lara Aknin et al. that reported that prosocial behaviour prompts a positive feedback loop with increased positive emotions [24] (p. 57). In short, prosocial behaviour seems to be inherently rewarding and increases positive emotions in the people engaging in it. These positive emotions draw a person’s attention to the needs of others which, in turn, make prosocial behaviour more likely.

In a similar vein, Frank Martela and Richard M. Ryan argued that human beings have inherent prosocial tendencies that directly contribute to their sense of well-being, and that benevolent acts are positively correlated with psychological need satisfaction [25]. The psychological needs outlined by Martela and Ryan are competence, relatedness, and autonomy. Prosocial behaviour fulfils these psychological needs in various ways: feeling effective in sustaining others contributes to competence and autonomy, and feeling connected (which often results from prosocial activities) fulfils the need for relatedness [25].

Further research suggests that being able to sustain others is an entirely different psychological experience from being able to sustain oneself. For instance, a study by S. Katherine Nelson et al. indicated that participants who engage in more altruistic behaviours experience more positive emotions and well-being than those who do not. In fact, those who engage in self-focused behaviour do not experience any increase in positive emotion [26] (p. 856).

A person’s *volition* to engage in prosocial behaviour is also positively correlated with increased well-being. A 2010 study by Netta Weinstein and Richard M. Ryan examined the degree of volition or autonomy behind a person’s altruistic behaviours and how that impacts upon the person’s experience. Weinstein and Ryan identified three motives that move people to perform altruistic behaviours: (1) for pursuit of personal gain or avoidance of loss, (2) the anticipation of self-esteem-relevant outcomes, and (3) a concern for the needs of others. Their results indicated that those altruistic behaviours that are undertaken more volitionally—or autonomously—are associated with the greatest increase in the person’s well-being [27] (p. 240). In short, the more autonomous the altruistic act is, the greater the associated well-being. They concluded that when people sustain others volitionally, they experience greater psychological-need satisfaction than otherwise, which in turn enhances their well-being.

A few studies have focused on the specific link between prosocial behaviour and loneliness, and incorporated loneliness scales into their assessments. A 2018 study by Carr et al. found that recently widowed older adults who engage in 2 or more hours of volunteer work per week report significantly less loneliness than do older widows who do not engage in volunteer activities [28] (p. 501). In response to these findings, the researchers concluded that engaging in volunteer work could potentially be a protective factor against experiences of loneliness. In contrast with other social activities, volunteering allows older widows to feel that they are making meaningful contributions while, at the same time, adding a sense of value to their lives (p. 502). Similarly, a 2021 study by Sunwoo Lee indicated that volunteering affects perceived social self-efficacy in older adults and that perceived control and self-competence have deep psychological effects strongly correlated with reduced feelings of loneliness [29]. The study noted that the experiences of loneliness in the sample population are largely mitigated by participating in volunteer activity because the participants are able to experience a sense of productivity.

While, in general, loneliness is associated with negative effects on prosocial behaviour [30,31], there are some data that indicate that this tendency varies according to gender and social situation. Two studies by Heqing Huang et al. found that while loneliness is negatively correlated with most kinds of prosocial tendencies, self-reporting lonely people are more likely to help others in public and this correlation is particularly strong among females [32]. For example, when asked to donate money to someone seeking funds for medical care, lonely participants are more likely to donate if told that they would be recognized as donors on the charitable foundation’s site. These findings support the *loneliness-reduction perspective*, which posits that lonely persons are theorized be motivated to repair and improve their social standing. Conversely, the *loneliness-perpetuation perspective* supposes that loneliness reduces sensitivity to the potential benefits of social situations that may satisfy the need to belong [33]. Some studies suggest that prosocial behaviour decreases specifically when a person encounters a social threat such as exclusion, bullying, or ostracism. Social exclusion jeopardizes fundamental human needs, such as belonging and self-esteem, as exemplified in a 2017 study by Oswald Kothgassner et al. [34] (p. 261). The study found that experiences of exclusion lead to social retaliation, by way of anti-social behaviour or aggression, and a consequently reduced frequency of prosocial behaviours such as helpfulness toward others (p. 267).

Finally, a few studies in the literature on prosociality suggest that it is a potential protective factor against loneliness for children and adolescents. A 2013 study by Emily Griese and Eric Buhs reported that engaging in prosocial behaviour significantly mitigates peer victimization and loneliness in fourth- and fifth-grade children [35] (p. 1052). Consequently, they concluded that prosocial behaviour can serve as a protective factor for those children who are being relationally victimized and suffering from loneliness. (The authors noted that these findings applied only to relational victimization and not to overt victimization (p. 1062).) Furthermore, a 2012 study by Susan Woodhouse et al., which examined loneliness and peer relationships among adolescents using the Adolescent Loneliness Scale (ALS), found that loneliness is inversely related to prosocial behaviour [31] (p. 273). The researchers hypothesized that this inverse relation may be the result of highly prosocial adolescents being more socially accepted and, therefore, less lonely. Another possibility is that lonely adolescents may be less inclined to engage in prosocial behaviour (p. 274–275). (There seems to be comparatively little research validating this scale. See [31] pp. 278–279.)

While these final few studies do utilise loneliness scales in their assessments, they do not include neededness items. Including such items in loneliness scales would not only make the scales more applicable to the growing body of research on prosociality and well-being but also align the quantitative work with emerging qualitative work.

## 4. Qualitative Studies on Neededness

In a 2020 study, Michelle Anne Parsons examined the relationship between being ‘unneeded’ and experiences of loneliness among older Muscovites in post-Soviet Russia. Through a series of interviews, Parsons hypothesized that social exchange practices are anthropologically connected to societal dimensions of loneliness [36] (p. 635). In other words, ‘Being unneeded in post-Soviet Russia is a culturally specific form of relational lack that suggests one possible way forward for an anthropology of loneliness, in which loneliness is connected to social exchange practices’ (p. 635). During the interviews, many older Muscovites used the expression ‘needed by nobody’ when discussing their experiences after the Soviet Union fell. This expression denotes that these civilians were no longer needed in their workplaces, by others, or by the state (p. 639). Parsons notes that among older Muscovites there seemed to be a recognition that being needed is a form of social protection (p. 640). Since being unneeded impacted upon their feelings of belonging and social integration, Parsons suggested that the inability to be needed may make a person more susceptible to loneliness (p. 644). Although none of these interviews claimed a direct link between their experiences of unneededness and the potential result of feeling lonely, instances of this relationship have been observed in other qualitative studies. One 2012 study by Judith Smith contained substantial references to the relationship between neededness and loneliness. For instance, one participant, Helen, described feeling lonely because ‘no one needed her anymore’ [37] (p. 301). Helen was recently widowed, all of her seven children had grown, and they were no longer dependent on her. Consequently, her experience of loneliness was mainly due to a recognition that she was no longer needed. Rose, another participant in the study, reported experiencing loneliness after her ill husband, whom she had cared for for the past 9 years, passed away. To mitigate her experience of loneliness, she opted to volunteer at the hospital because she felt that contributing to the well-being of others alleviated her loneliness. She even stated in the interview, ‘You have to be needed. If you don’t feel like you’re needed, then you’re in trouble’ (p. 303). Smith noted that throughout the series of interviews, coping patterns emerged amongst lonely participants. Notably, one of the most prominent coping methods for dealing with loneliness was helping those in need such as engaging in volunteer work (p. 299). Thus, this qualitative study suggested that people *do* recognize how their neededness impacts upon their feelings of loneliness and, hence, may seek out opportunities to feel more needed.

Although most of the qualitative work focuses on older adults, there is some evidence for the unneededness-loneliness association in young adults. A 2020 study by Chikako Ozawa-de Silva interviewed Japanese college students to assess their experiences of feeling needed and found a strong connection to loneliness [38] (p. 630). Ozawa-de Silva initiated an examination of Japanese students posting on suicide websites, finding that they express severe feelings of loneliness associated with feelings of not being needed (p. 624). More precisely, frequent expressions on these suicide websites signalled ‘(1) severe loneliness, to the point of feeling too lonely to die alone and wanting others with whom to die; (2) an absence of any reason to keep on living, any sense of what life is for, or any meaning in life; and (3) a feeling of not being needed and an absence of social connections and sense of belonging’ (p. 624). Consequently, Ozawa-de Silva hypothesized that the need to be needed could also be considered as a manifestation of the fear of loneliness (p. 623). In response to the findings on the suicide websites, Ozawa-de Silva posed particular questions relating to neededness in the interviews: (1) Have you ever felt that you are needed? and (2) Is such a feeling important for you in your life? Responses varied, from some students saying that they would like to be needed by others to other students saying that being needed is the meaning of life (p. 626–629). In response to both the expressions on the suicide websites and the interview responses, Ozawa-de Silva concluded that if people are able to be needed by others their experiences of loneliness might be mitigated (p. 630).

In addition to better aligning loneliness scales with other quantitative and qualitative studies, adding *neededness* items to loneliness scales would improve them in at least three specific ways: (1) augment their capacity to pick out influencing factors for loneliness, (2) better understand the emotional tone—and the variation in tone—of persons’ experiences of loneliness, and (3) identify suitable interventions to alleviate loneliness. The next section explores the first two of these rationales. Section 7 explores the third.

## 5. Ways That Neededness Items Would Improve Loneliness Scales

### 5.1. Understanding the Risk Profile for Loneliness

Including neededness items would improve loneliness scales by increasing their capacity both to identify distinctive risk factors for loneliness and to understand whether (lack of) neededness informs some of the recognised risk factors. Recognised risk factors for social isolation and loneliness include 1. psychological factors, 2. physical factors, 3. social factors, 4. time-use factors, and 5. circumstantial factors, a set of influences that gives only an implicit nod to neededness [3].

1. Psychological factors of note include the personality traits of either intraversion or extraversion and the presence or absence of a history of mental health conditions. Whereas extraversion—i.e., the tendency to be outgoing, talkative, and energetic—is regarded in the psychology literature as one of the main protective factors against loneliness, intraversion—i.e., the tendency to be reserved, reflective, and depleted by interaction—is part of the risk profile for loneliness. Extraversion not only protects against the onset of loneliness but can mitigate its effects in various ways since extraverts are more likely to find ways to connect with others and to expand their social circles [39]. Extraverts also tend to report a higher quality of social relationships than intraverts do, are more satisfied in their social relationships, and have larger social networks [40] (p. 2). (Note that studies [39,40] were conducted during the COVID-19 pandemic and the findings may be context-specific.) A possible explanation for the protective value of extraversion focuses not on whether intraverts are more inclined toward feeling isolated irrespective of their circumstances, but instead on the kinds of social relationships that intraverts need to have to feel connected. Intraverts tend to favour fewer, but deeper connections, which means that potentially only a narrow range of social options will suffice to meet their social needs. Whereas the extravert might find sufficient social support through both shallow and deep connections, the intravert may require deep connections to feel socially connected. 

Although, in general, extraverts are less likely than intraverts are to be lonely, they are more susceptible to loneliness if they experience a drastic change in their social network. This is likely because they experience a larger deficit (or a larger ‘gap’) in their social provisions from what they are used to. Essentially, the more social resources a person has, the more they have to lose and, thus, extraverts will likely feel this shift more dramatically than do their intraverted counterparts [40] (p. 9).

Other psychological factors that correlate with loneliness are low levels of conscientiousness and agreeableness. According to a 2020 study by Cooper et al., people who are high in conscientiousness are more likely to seek out and engage in social tasks [39] (p. 22). In terms of agreeableness, this same study finds that participants who report lower agreeableness have worse relationships with their friends and family members than do those who report higher agreeableness. This study concludes that agreeableness is positively correlated with sociality, higher quality of relationships, and higher relationship satisfaction (p. 23). A contrasting perspective from psychopathology would note that agreeableness and conscientiousness can pull apart: people can find themselves in an agreeableness/conscientiousness double-bind. For example, a person who is experiencing a migraine might cancel a work meeting (low conscientiousness) or they might attend the meeting and come across as grumpy (low agreeableness).

Each of these psychological factors potentially interacts with, and is compounded by, low levels of neededness. Intravert tendencies and lower levels of agreeableness or conscientiousness may pose greater risks for loneliness when a person has inadequate ways to see themself as supporting others and as being important to others. In short, the few efforts they might be able to make may well come to naught.

2. Physical factors that typically predict higher rates of loneliness include: reduced capacity with increased age, mental health conditions, and physical disabilities [41] (p. 388). The extent of a person’s physical abilities can predict loneliness because socializing often requires a great deal of physicality. Functional and fine-motor impairment is considered a risk factor for loneliness because it limits a person’s ability to dress themself, feed themself and utilize money, all of which are imperative to social interaction in the public sphere (p. 393). These experiences also correlate with a sense of lost independence, increased psychological stress, and—crucially for present purposes—feelings of being unable to participate. Fine-grained studies of physical impairment and loneliness could drill down into the underlying reasons for such feelings, which may often be rooted in the person’s sense that they are unable to contribute to others’ well-being and, hence, are unable to be needed by them.

Chronic illness can also further impair function and reduce a person’s ability to socialize. People who have chronic health issues generally report lower satisfaction within their interpersonal relationships [42] (p. 1190). However, these effects can be mitigated with high-quality healthcare and high levels of family support. Section 7 below notes ways in which interactional contexts might be designed to increase persons’ opportunities to enjoy horizontal connections (i.e., symmetrical or mutually supportive connections) rather than vertical connections (i.e., asymmetrical or one-way-support connections) in which they are largely passive recipients rather than active, needed participants.

3. Social factors that correlate with increased loneliness include non-married status, widowhood, lack of religious affiliation, lower quantity of social relationships, lower quality of social relationships, and absence of close interpersonal relationships [41] (p. 388). Typically, married people report less loneliness than do non-married people. Of course, many married people experience loneliness, which suggests that marital loneliness is at least partially dependent on the quality of the relationship. (The presence of children is a complicating factor.) People who are married and report high levels of loneliness may do so as a result of the absence of other relationships or because this key relationship is wanting. A study by Ning Hsieh and Louise Hawkley (2018) indicated that older spouses in aversive or indifferent marriages are lonelier than supportively married counterparts, and that effects of poor marital quality on loneliness are not ameliorated by good relationships with friends and relatives. So, while marriage generally reduces levels of loneliness, the kind and quality of the marriage matters. ‘For instance, past research has shown that marriage is associated with reduced loneliness only to the extent that the marital partner serves as a confidant’ [43] (p. 1320). These findings would support the view that neededness is operative in the background, on the assumption that both being needed and valued by one’s spouse and needing and valuing one’s spouse are key components of a supportive marriage.

Married couples also typically cohabitate, and living with others is thought to be a significant way to prevent or lessen loneliness. A 2009 study by Gerdt Sundstrom et al. found that, for older adults, living with a partner or spouse is associated with lower levels of loneliness than living alone [44] (p. 273). In fact, this study indicated that all other types of living arrangements are associated with a lower prevalence of loneliness when compared to living alone. In line with this trend, married people who report that their spouse is more active and present at home experience higher relationship quality and lower levels of loneliness [41] (p. 393). Furthermore, participants who perceive greater importance in their relationships typically report less loneliness, have more friends, and have higher contact rates with their social networks [42] (p. 1191). In sum, a broad consensus in the literature reports that protective social factors include both quantity and quality of interpersonal relationships, i.e., connections that notably tend to supply a person with ready ways to serve someone else’s well-being and, thereby, to feel important or necessary to that person.

4. Relevant time-use factors in the loneliness risk profile include how a person uses their leisure time. Some studies identify a negative correlation between well-being and spending one’s leisure time alone [45]. Others identify a correlation between spending considerable amounts of time on social media and feeling lonely. Although small amounts of social media seem to be a protective factor [3], a 2018 study by Melissa Hunt et al. found that college students who limit their social media use to 30 min or less a day, with no more than 10 min per day on a given platform, were less lonely and depressed over the course of several weeks than a control group that maintained their normal use [45,46]. Such findings do not straightforwardly align with the view that unneededness is a key influencing factor for loneliness since—superficially at least—those students spending more time on social media are engaged in a sociable enterprise. The task is to determine how often such social media activity is aimed at supporting others’ well-being and whether that activity can have approximately the same protective value as offline equivalents.

5. Circumstantial factors that form part of the risk profile for loneliness include demographic factors such as older age or indeed young age, as well as experiencing life transitions or being a member of a vulnerable group such as those in poverty, financial insecurity, food insecurity, low education, and a lower socio-economic status [3,47]. Although such factors are unsurprising elements in a risk profile for loneliness irrespective of a link to lack of neededness, nevertheless it is worth noting that one significant effect of being financially insecure is that this diminishes a person’s capacity to be hospitable toward others [48].

### 5.2. Understanding the Emotional Tone of Loneliness

A second way that neededness items could improve loneliness scales is by refining their capacity to describe the emotional tone or felt quality of loneliness. Our speculative proposal is that the underlying cause or influencing factors behind a person’s loneliness will inform the quality of their experience of loneliness and especially its emotional tone. The extant literature on loneliness dedicates comparatively little attention to emotional tone (probably for the good reason that persons vary in how they respond to situations), and the descriptive language used to characterise loneliness as an experience is often ambiguous. Whereas contemporary accounts describe loneliness as having a negative or unpleasant tone, older accounts characterized loneliness neutrally as *aloneness* or *solitude*, which can have a pleasant, neutral or unpleasant tone [49]. Contemporary accounts differ from each other insofar as some treat loneliness as an emotion itself, and some treat it as associated with or characterized by certain emotions and psychological features. In general, if loneliness is described as a *distressing feeling* arising from unfulfilled social needs [13] or as a cognitive discrepancy between needed and actual relationships [50], then it is treated as including a set of likely emotional responses including feeling unhappy, unloved, empty, restless, despondent, or depressed. One question worthy of empirical study is whether a person might cognitively perceive a gap between their actual and desired relationships, but not feel distressed by it. The emotional tone of loneliness may be influenced by the extent to which a person identifies with their loneliness as part of their temperament, in the same way that Charles Spielberger’s well-established anger scale distinguishes, in different subscales, *anger as a trait or temperament* from *anger as a reactive state* (captured by statements like ‘I am a hot-headed person’ in contrast with ‘It makes me furious when I am criticized in front of others’) [51]. (In the case of anger, the variability in emotional quality can include resentment, humiliation, guilt, and tenseness in differing degrees. Theories of emotion that define *emotions* as bodily feelings, i.e., anger is constituted by a rise in heartrate, sweating, also permit these distinctions—for example, some instances of anger reactions include a churning feeling in the stomach, but others do not.)

Pertinent to the present paper is the question of whether and how the emotional aspects of loneliness might manifest their specific sources, i.e., whether the tone of a person’s loneliness varies according to its cause and is a marker for that cause. For instance, if the cause of someone’s loneliness were being unneeded or lacking adequate ways to contribute to others’ well-being, then perhaps that person’s emotional responses would centre around feeling unwanted or unwelcome or even feeling despair and worthlessness. If, by contrast, their feelings of isolation were largely the after-effects of angry outbursts, then perhaps their response would centre around feelings of being under threat or, ultimately, fear. As we note above, these proposals are speculative. It is possible that a person who has displayed anger may then feel unwanted or unwelcome rather than under threat, or they may evaluate themself as having acted in a threatening way, leading to feelings of guilt and worthlessness. Equally, someone who perceives themself as unneeded might become scared and feel under threat as a result, as society seems not to value people like them. Given, though, that some feelings are less commonly understood as constitutive of loneliness responses, some instances of loneliness may go unrecognized and undetected unless the complete profile of loneliness—and the relevance of neededness—is fully understood. Indeed, since it is possible to be lonely without being fully aware of it, loneliness scales must be designed to capture the full range of persons’ experiences and responses.

The emotional contours of loneliness can be understood in terms of appraisal frameworks, which attribute differences in persons’ emotional responses to their individual interpretation or *appraisal* of a given inciting situation (and not to facts about the situation *per se*). Research on emotions with a cognitive orientation typically accepts this assumption about the role of appraisals [52]. Dimensions of appraisal include the importance of the situation, its expectedness, the person(s) responsible for the situation, and the degree of control the person has over the situation [53]. As this suggests, appraisals are imbued with individual values; the quality of a person’s loneliness triggered by a failed social connection, for example, can reveal the extent to which that social connection matters to them.

It is plausible that the experience of loneliness as characterized by feeling unneeded is qualitatively distinct from loneliness caused by other kinds of unfulfilled social needs such as needs for nurturing care, acts of service, gentle touch, etc., as they correspond to different appraisal dimensions. While existing studies sometimes confound a person’s appraisal of a situation and the situation itself (and, hence, render it difficult to evaluate whether appraisals are sufficient for emotional responses), both interpretations allow for the emotional tone of loneliness to depend on its cause [54]. If it is the person’s appraisal that matters, then the quality of their loneliness provoked after being excluded from some social event might depend on whether they perceived the act of exclusion as deliberate, how socially valuable they judged the event to be, whether they based their sense of self-worth on events of that kind, and so on. For example, if the event was professional, the emotional tone of their loneliness might lean towards restlessness. If, by contrast, the event was romantic, their feelings might lean more towards lovelessness. Their failure to contribute meaningfully to their social surroundings could potentially lead to feelings such as despair over their perception that their presence is unwanted, frustration over their activities being trivial, and panic over their inability to be useful, all of which appear to differ from other sources and experiences of loneliness, such as stress over workplace isolation, or coldness over romantic rejection.

Incorporating neededness items into loneliness scales is necessary not only to develop a more accurate understanding of the psychology of loneliness as the above discussion shows, but also to determine as a matter of practical ethics and politics whether and how to prioritize persons’ interests in neededness. The next section briefly outlines the theoretical case for taking neededness seriously in ethics and politics.

## 6. The Human Need to Be Needed

As the studies cited above indicate, it is species-typical of human beings to have an inherent capacity to connect socially and to experience anxiety when isolated. In Matthew Lieberman’s words, ‘being socially connected is a need with a capital N’. He wrote that: ‘Maslow had it wrong. To get it right, we have to move social needs to the bottom of his pyramid. Food, water, and shelter are not the most basic needs for an infant. Instead, being socially connected and cared for is paramount. …our biology is built to thirst for connection because it is linked to our most basic survival needs’ [55] (p. 43).

Nevertheless, as noted above, persons differ in their levels of vulnerability to social disconnection and loneliness due to a combination of physical, psychological, and environmental factors, which can inform both their abilities to self-regulate when confronted with isolation and their social cognition, i.e., how they perceive others [7] (p. 14). These varying abilities and vulnerabilities can influence a person’s experience of transient loneliness as it devolves into chronic loneliness. If a person’s social cognition predisposes them to perceive those around them as frightening, this perception can become a negative expectation, leading them to develop a pre-emptively defensive posture that can compromise their ability to self-regulate [7] (p. 16). These factors lend themselves to the experience of chronic loneliness. A person’s biology and environment also shape their level of social need. In relationships where each participant has different or conflicting levels of social need, one (or perhaps each) person can come to be isolated even within the social relationship because of the mismatch in their levels of need [7] (p. 17).

An important conceptual distinction was noted at the outset between *contingent needs* and *non-contingent needs*. Whereas contingent needs depend on changeable conditions such as a person’s desires, non-contingent needs are needs that a being necessarily has as the kind of being it is. This conceptual distinction reveals a further, morally evaluative distinction: non-contingent needs pertaining to a being’s survival have special moral importance. Soran Reader and Gillian Brock hold that a person’s non-contingent, morally urgent needs include agency, flourishing, and avoidance of harm, because a person who fails to achieve these things cannot maintain a human life; they might persist as an animal, but not lead a *human* life. Put differently, securing these needs is necessary for them to live a minimally good human life [5,56].

Drawing on empirical evidence pertaining to sociality and loneliness, Kimberley Brownlee adds to the list of non-contingent, morally urgent human needs several social needs that are essential to a minimally good human life. One of these is the *need to be needed* or, more precisely, the need to have the social resources (i.e., the social abilities, opportunities, and connections) necessary to undertake meaningfully to support others’ well-being. This need to be needed plays an essential role in human development and flourishing, as it is bound up with the need to belong [6] (chapters 1, 3) [57] (p. 497).

Brownlee maintains that the need to be needed is motivated by three factors (1) self-interest; (2) other-interest (i.e., interest in others’ flourishing); and (3) community-interest (i.e., interest in the well-being of larger communities and societies). Concerning self-interest, when a person makes meaningful contributions to others’ well-being, they create a form of insurance for themself by increasing the chance that they will be valued by others, if not indispensable to them. Concerning other-interest, when a person is meaningfully useful to others, they can indirectly provide some insurance to their dependents who may not themselves be useful to others. Finally, concerning community-interest, when a person is useful to others, they add their social labour to the collective effort to secure all members’ social and material interests.

The motivations to be important to others and to be able to serve others’ well-being tie in closely with feelings of mattering. Morris Rosenberg and Claire McCullough held that *mattering* concerns a person’s need to feel significant to others, which is based partly on a knowledge and awareness of meaningful connections and strong relationships [58]. Being significant to others, they noted, involves those others noticing the person, perceiving them as significant, and relying on that person. By contrast, Isaac Prilleltensky postulated that mattering is a conjunction of feeling valued and adding value (i.e., contributing meaningfully to others’ well-being), and that in order to fulfil a need for mattering, a person must achieve both things [59].

There are many ways in which a person can be useful to or valued by others, but not all of those ways are equally efficacious in satisfying that person’s interests in *neededness*. What matters—for the study of loneliness—is to classify those various ways and to determine which ways (if any) are reliable correlates of either social connectedness or loneliness.

Briefly, the ways in which a person can be needed can be specified in relation to at least four continua that we take to be evident upon theoretical reflection and observation of indicative examples: a person can be needed in ways that are (1) trivial, at one end of the spectrum, or significant, at the other, (2) instrumental or non-instrumental, (3) momentary or persistent, and (4) fungible or non-fungible. For instance, a taxi driver is needed in a relatively trivial, instrumental, momentary, and fungible way by the unhurried tourist who wants to get to the airport. Typically, one taxi driver is just as good as any other in the local area, and the driver knows this. By contrast, typically, a mother or father is needed by their young child—and that child needed by their parent—in highly significant, non-instrumental, persistent, and non-fungible ways: no one else will do. A study on bipolar disorder further underlines the value in being needed in meaningful if not irreplaceable ways; the point made in this study applies *mutatis mutandis* to loneliness and aligns with the qualitative studies summarized in Section 4: ‘Being needed is an experience, profoundly affecting life with BD [bipolar disorder]. It means having a strong reason to continue the particular, daily struggle that characterizes life with BD. Being a father or mother with BD means among other things having a reason, i.e., the children, to take responsibility for oneself and to be open for and consider dependency. Being needed by one’s children may make it possible to make breakfast for the children even though the whole being wants to stay in bed and sleep, trying to escape hopelessness, but also to stay focused in an emotional chaos…Having children and being needed by one’s children may also prevent one from giving up the reason to live by committing suicide’ [60] (p. 4653).

There are many ways to interpret the claim that being needed gives a person a reason to take care of themself. On one interpretation, the person sees themself as instrumentally useful—a tool—to serve others and, hence, they are motivated to take care of themself purely to benefit others (self-sacrificing altruism). On another, more egoistic interpretation, they see helping others as a way to help themself to take better care of themself and, hence, they are motivated to serve others in order ultimately to heal themself. On a third, there is a harmony between the person’s motivations; they both aim at others’ well-being and recognize the value for themself in being needed by those others. One question worthy of empirical study is which, if either, motivation is the primary one: in other words, which motivation exercises a veto. For instance, would this person, who is prone to loneliness, abandon their role as a needed parent if being needed ceased to help them to feel better? Two more general questions ripe for empirical study are: (1) To what extent is a person’s perception that they are needed by others influenced by the significance, persistence, non-instrumentality, and non-fungibility of the ways in which they are needed? and (2) To what extent does the protective effect of their being needed depend on the specific ways in which they are needed? The next section reflects on these questions theoretically, offering a taxonomy of social-interaction contexts and association types as better or worse sites in which to satisfy interests in neededness.

## 7. Implications for Public Policy and Treatment Design

### 7.1. Social Interaction and Proto-Associative Support

Which kinds of associations and interactional contexts give a person adequate chances to engage in prosocial behaviour, be useful to others, and feel needed? Specifying the interactional dynamics of different association types and non-associative contexts can reveal, in general terms, the quantity and quality of neededness opportunities that a person in those contexts enjoys.

Although most people interact regularly with non-associates, in brief encounters with strangers (e.g., a passing smile, asking for the time, helping someone to retrieve fallen groceries) and in exchanges with acquaintances whom they may see frequently but never know well, nevertheless most of a person’s chances to feel important to others or to further others’ well-being arise within their existing associations. The kinds of chances they have depend, therefore, on the types of associations they enjoy. Common association types include not only romantic, familial, and spousal, but also fraternal, collegial, political, creedal, professional, medical, transactional, recreational, and philanthropic. Whereas some association types are hierarchical or vertical by nature, such as employer–employee and parent–child associations, others are paradigmatically egalitarian or horizontal, such as friendship. Whereas some associations are formal, such as doctor–patient relationships, others are informal or intimate, such as spousal relationships. Whereas some are friendly, others are antagonistic. When two people share a connection that falls under more than one association type, e.g., two spouses are also work colleagues or a doctor and a patient become friends, the norms and purposes that define each association type can clash and sometimes lead those people to abandon one association type in order to preserve the other: the patient might find a new doctor or one spouse might move to a different employer. 

An association type and its surrounding social norms strongly influence the ways that the parties involved can interact and support each other. This is partly because association types differ in their official purposes. Some associations are officially defined as one-way or two-way service provision, some as commercial transactions, some as pure companionship, and so on. A given association type can have multiple official purposes, which can pull in different directions, but those official purposes determine what space participants have—officially—to be needed within the association, and thereby can either support or hinder participants in being unofficially needed. For instance, in visits to the doctor, only the service provider is the officially needed party, and the specific structure that defines doctor–patient relations leaves little space for the patient to present themself as unofficially socially necessary to the doctor. (Both participants know that the patient is instrumentally useful to the doctor’s finances, and it is a question worth exploring whether being needed in that way could secure a sense of social connection. The hypothesis to test would be that it cannot.) By contrast, other service-oriented associations are less formal and, hence, leave more space for the service receiver to feel unofficially useful. For example, although, officially, the Spanish tutor gives her tutee a Spanish lesson, the masseur gives a client a massage, and the social care worker visits a care receiver, nevertheless there is—conventionally—sufficient informal space within these association types for the tutee, client, and care receiver to feel important within the association, provided that certain background conditions are met. For example, when the same social care worker visits the care receiver each week, the receiver can then have a decent chance to develop a joint narrative with the carer, be a witness to the carer’s life, show that they are worthy of trust, and so on. When the social care service is provided by a different person each week, however, the care receiver is robbed of all opportunities to show that they can be useful and important within this association [6] (chapter 3) [61].

Participants’ attitudes toward each other also play a key role in affirming or refuting their perception that they are needed within a given association. Participants’ attitudes are particularly relevant in social settings that are otherwise coloured by a transactional dimension, e.g., paying for a therapy session, paying for sex, or paying for social company, versus interactions that are overtly non-transactional, e.g., taking part in an informal book club or chatting with family members online (where no one has an ulterior motive), or being intimate with a longstanding loving partner. One hypothesis to test is whether, and when, the transactional element undermines a person’s sense of neededness and when it is compatible with retaining a sense of neededness. A further hypothesis to test is whether having sufficiently many cordial interactions with strangers or acquaintances—including commercial social transactions—could give a person a well-being-protective level of neededness, especially in the case where the person lacks closer social ties. Non-associative interactions do serve a distinctive social function in signalling to a person that they are safe and generally accepted within the wider community, and such interactions are also often proto-associative: if two strangers interact regularly, they tend to begin to lay down a foundation together for an association to form between them [6] (chapters. 4–6).

### 7.2. The Design of Appropriate Remedies

If the above analysis is corroborated by fine-grained empirical testing of persons’ perceptions of their neededness, then various things would follow for healthcare and public policy. Briefly, if being needed, or having good grounds to perceive oneself as needed, is crucial to staving off or overcoming loneliness in specific domains (or overall), and if only some association types and interactional contexts provide adequate space for persons to be needed in meaningful ways, then healthcare professionals and policymakers must attend to whether persons have sufficient opportunities to interact and associate in ways that support their interests in neededness. This is particularly true for people who require some assistance to ensure that they have persistent access to adequate social opportunities, such as young children, older adults, and persons with severe disabilities. The people responsible for securing their persistent access to social inclusion must attend to whether the association types and interactional contexts in which they participate provide them with adequate chances to be needed or merely ensure they are not constantly alone. A hypothesis to be tested is whether a person might feel lonelier if they are socially included but made to feel wholly dependent, passive, and non-contributory, than if they are left largely alone.

Additionally, with regard to persons who are broadly able to secure their own social needs, policymakers have reasons to attend to the dimensions of urban planning and community funding that foster or hinder persons’ efforts to interact in ways that enable them to be needed.

Finally, understanding which association types serve or thwart a person’s sense that they are needed can yield greater understanding of the benefits and harms of surrogate or virtual alternatives. To what extent can feelings of being needed be generated through online forms of interaction? To what extent do the features noted above, of hierarchicality, formality, reciprocity, passivity, etc., influence a person’s sense of neededness within mediated, online, virtual, robotic, or animal variants of traditional association types? One hypothesis to test is that, when it comes to providing space in which to satisfy interests in neededness, online social settings display broadly similar profiles to their offline counterparts but are generally poorer forums in which to satisfy the need to be needed.

## 8. Conclusions

Despite the evidence that neededness is a protective factor for well-being, prominent tools for measuring loneliness—the loneliness scales—fail to properly incorporate items that assess either a person’s chances to serve others’ well-being or their perception of whether they are needed by others. The only neededness-focused items in existing loneliness scales are confined to romantic subscales. This is insufficient because neededness plays a key role in friendships, family relationships, and large community bonds as well.

This paper has shown how loneliness scales that measure neededness could (1) more accurately align with, and be useful to, the growing body of evidence showing that feeling needed or not is crucial to the story about the nature of, and conditions for, well-being, (2) provide a richer understanding of recognized risk factors for loneliness and possibly identify further, under-explored risk factors, (3) offer greater insight into the quality or emotional tone of loneliness by better specifying differences in persons’ experiences, and (4) through this more fine-grained understanding, inform the design of appropriate remedies and preventive strategies including, notably, in urban planning and public social-interaction design.

Avenues for further research include not only the study of the various hypotheses proposed above, but also an examination of whether the two neededness factors discussed here—being useful and being valued—actually operate in isolation. Isolating these items could further refine metrics of loneliness and clarify the relationship between different dimensions of neededness. It would be noteworthy, for example, if a person were to report that they *do* feel needed, e.g., they have loved ones who they know rely on them, but they nonetheless find that in all or most other respects they feel lonely. Such cases may be common: for example, a woman who is pressured to have a family and be a stay-at-home mother may feel profoundly needed by her dependents but socially isolated from friends and colleagues, and lonely as a result. Or a sex worker may feel highly needed by clients but resentful of these relationships, and lonely as a result. Indeed, the sex worker may well experience loneliness as a result of feeling needed in ways that she herself does not value, in addition to not having chosen them freely. It would be noteworthy too, if a person were to report feeling needed, but unable to contribute in good ways to those others’ well-being, or if they felt able to contribute meaningfully to others’ well-being but nonetheless felt unneeded. For example, a powerful business executive could know that she is valuable in the company but lack a sense that her day-to-day activities meaningfully improve the lives of others. Conversely, a nurse working difficult shifts could contribute crucially to the health of others but feel that his position in the work environment is easily replaceable and therefore trivial. Each scenario could plausibly set the scene for loneliness, but for different reasons and, presumably, with different attendant emotions. If such neededness-related experiences of loneliness continue to go underreported, future research will fail to capture a critical element of loneliness.

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
