# Peer review of "The Missing Measure of Loneliness: A Case for Including Neededness in Loneliness Scales"

_ijerph, 2021, doi:10.3390/ijerph19010429_

Round 1
Reviewer 1 Report
This is a well-written review article that proposes that neededness should be included in loneliness scale. This is an interesting proposition, however, the relationship between loneliness and altruistic behaviours is not well-evidenced yet in the existing literature. The authors make use of some wider literature on prosocial behaviour to support their arguments. Arguments are made clearly and concisely throughout, although at times arguments are not always well-reasoned. The article is well-written, with a good structure and a lovely flow that made reading/reviewing a pleasure. It is an interesting article that will make important contributions to the field and dialogue relating to conceptualisation of loneliness and how it is measures.
Despite this I do have some comments for revision:
The article is quite focussed around adult prosociality and loneliness literature and adult loneliness scales and I wonder if "neededness" is as important for children and adolescents. Some discussion or acknowledgement of this in the document would be good (perhaps in the limitations section/future directions).
The review does focus on prosociality literature but there is some literature that examines helping and volunteering and impacts on loneliness which would have been good to see mentioned here in this review. I suspect that is because these do not measure neededness per se - but given that there is not a strong evidence base for whether helping and volunteering reduces loneliness - there needs some mention of this literature in this review. e.g.
Carr, D. C., Kail, B. L., Matz-Costa, C., & Shavit, Y. Z. (2018). Does becoming a volunteer attenuate loneliness among recently widowed older adults?. The Journals of Gerontology: Series B, 73(3), 501-510.
Also worth considering some of the studies relating to loneliness and prosocial behaviour e.g.
Huang, H., Liu, Y., & Liu, X. (2016). Does loneliness necessarily lead to a decrease in prosocial behavior? The roles of gender and situation. Frontiers in psychology, 7, 1388.
Kothgassner, O. D., Griesinger, M., Kettner, K., Wayan, K., Völkl-Kernstock, S., Hlavacs, H., ... & Felnhofer, A. (2017). Real-life prosocial behavior decreases after being socially excluded by avatars, not agents. Computers in human behavior, 70, 261-269.
On line 203 the following comment is made:
"Notably, none of these studies utilises loneliness scales in its assessments"
but the study cited immediately before this (Lee, S. (2021). Volunteering and loneliness in older adults: A parallel mediation model. Aging & Mental Health, 1-8.) did use a loneliness measure.
On line 212 the authors note the following factors influencing loneliness: psychological factors, social factors, time use, physical factors, and circumstantial factors, they then set out to describe each of them in the following section but mention only psychological and social factors. I also think that there is a missed opportunity here to explicitly related each of these influencing factors to neededness to strengthen their argument that neededness is an essential component of the loneliness experience/concept.
There is a complete lack of discussion of marginalisation, discrimination and social exclusion in relation to the experience of loneliness and a strong focus in the paper on individual behaviour as a cause of loneliness. Importantly loneliness is experienced when people are discriminated and social excluded.
on lines 272 to 273 a study is mentioned which was conducted during the COVID-19 restrictions where people were particularly restricted from meeting their social needs if they were living alone as they were not able to socialise away from their home - this may mean that this loneliness is overinflated in this study and is therefore misrepresented in the document without acknowledgement in the text that it is during covid. Importantly single people may not typically experience this level of loneliness outside of COVID times when they are able to supplement the lonely experience of living alone with social activities. I think it is important to acknowledge this in the text or use another example.
The argument in the text that including neededness would help reflect the emotional tone of loneliness is not well argued. Indeed the authors have included a footnote about the speculation here. It is not possible to infer a persons emotions in relation to a given circumstance as people respond differently to a given situation. I think some discussion of this is needed in the text if this rationale is used for including neededness in a loneliness measure rather than just adding a foot note.
The authors do give other rationales for this which have stronger arguments/rationale, e.g. to identify cause of loneliness to suggest interventions - this is briefly detailed at line 341. This final argument here seems more relevant to me - that understanding someones loneliness as related to neededness helps us to establish the best course of action to support reductions in loneliness i.e. by suggesting vonuteering or helping activities in the case of neededness but cognitive behaviour therapy perhaps in the case of loneliness that related to a sense of threat and more linked to social anxiety for example.
The authors clarify the 4 reasons for including neededness in loneliness measures at line 540 - this is useful as a framework for the paper perhaps. I don't feel that all these reasons are sufficiently discussed in the paper earlier so a review of the paper to ensure all these reasons are discussed and/or a restructure to use these arguments as a framework is advised. (alongside some consideration of how to revise the discussion the more speculative one about emotional tone of loneliness in the text).
Reviewer 2 Report
Thank you for the opportunity to review this work. This is an interesting piece of work and I commend the authors for critically reviewing the existing conceptualisation and measurement of loneliness. I hope the authors find the following comments useful.
Generally, there is some empirical evidence to support the arguments made, but there are also a lot of places that lack sufficient justification. Additionally, what would strengthen this work, but is currently missing, is evidence from qualitative work. Beyond the theory, do individuals actually report that need is an important aspect of loneliness? (see also comment 8)
- Introduction: could you provide some references when describing contingent vs. non-contingent needs?
- Overview of existing loneliness scales: Where the authors say "This 20-item, unidimensional scale". The UCLA was initially developed to be unidimensional but there is no sufficient support to support unidimensionality. So I would remove "unidimensional"
- "In response to this new dichotomy between social and emotional loneliness, researchers have sought to develop new loneliness measures": this is also the case for existing measures like the UCLA. there have been some efforts into exploring a multidimensional structure (see Hawkley and colleagues work). I suggest adding this in.
- "only a person’s perceptions of her..". I realise that in different languages "person" is gendered (either female or male) but I'd suggest changing "her" to either "his/her" or to "their".
- Can you support the following with appropriate evidence? "Typically, married people report less loneliness than do non-married people. Of
course, many married people experience loneliness, which suggests that marital loneliness is at least partially dependent on the quality of the relationship. People who are married and report high levels of loneliness may do so as a result of the absence of social relationships" - ". Furthermore, subjects" - I would avoid using the term "subjects" and instead use participants or individuals
- can you clarify if the following is based on evidence, or the authors' view? "Briefly, the ways in which a person can be needed can be specified in relation to at least four continua"
- I appreciate that an example of an interview was provided, but I find it difficult to see how BD is related to the concept of loneliness. Given especially that the majority of measures considered in the manuscript are generic measures of loneliness to be used in the general population. What would really strengthen the paper is the consideration of qualitative work that has explored the meaning and experiences of loneliness. If "need" doesn't come up in such work, then how can the authors explain this? It is extremely important to consider theories in the development of measures, but without focus groups and without the consideration of what users feel loneliness is, we will not be able to develop robust measures. If such themes have not come up, is it because the theories are not accurate, or because we haven't been asking the right questions? These issues should be considered and discussed.
